# Intestinal Mucosal Barrier Improvement with Prebiotics: Histological Evaluation of Longish Glucomannan Hydrolysates-Induced Innate T Lymphocyte Activities in Mice

**DOI:** 10.3390/nu14112220

**Published:** 2022-05-26

**Authors:** Shih-Chang Chang, Hui-Hsun Chiang, Chih-Yi Liu, Yu-Ju Li, Chung-Lun Lu, Yung-Pin Lee, Chi-Jung Huang, Ching-Long Lai

**Affiliations:** 1Division of Colorectal Surgery, Department of Surgery, Cathay General Hospital, Taipei 106438, Taiwan; cgh06719@cgh.org.tw; 2School of Nursing, National Defense Medical Center, Taipei 114201, Taiwan; huihsunchiang@mail.ndmctsgh.edu.tw; 3Division of Pathology, Sijhih Cathay General Hospital, New Taipei City 221037, Taiwan; cyliu@cgh.org.com; 4School of Medicine, College of Medicine, Fu Jen Catholic University, New Taipei City 242062, Taiwan; 5Division of Cardiology, Department of Internal Medicine, Kaohsiung Medical University Hospital, Kaohsiung 80756, Taiwan; 1040647@kmuh.org.tw; 6Aquatic Technology Research Center, Agricultural Technology Research Institute, Xiangshan, Hsinchu 300110, Taiwan; 1041002@mail.atri.org.tw; 7Research and Development, Healthy-Bioceuticals Company, Taipei 114201, Taiwan; yplee@healthy-bioceuticals.com; 8Department of Medical Research, Cathay General Hospital, Taipei 106438, Taiwan; aaronhuang@cgh.org.tw; 9Department of Biochemistry, National Defense Medical Center, Taipei 114201, Taiwan; 10Division of Basic Medical Sciences, Department of Nursing, Chang Gung University of Science and Technology, Taoyuan 333324, Taiwan; 11Research Center for Chinese Herbal Medicine, Chang Gung University of Science and Technology, Taoyuan 333324, Taiwan

**Keywords:** prebiotics, glucomannan, T-cell activation, lymphoid aggregates, colitis

## Abstract

Use of prebiotics is a growing topic in healthcare. A lightweight molecule and water-soluble fiber ingredient, longish glucomannan hydrolysates (LGH), has been developed to improve the intestinal mucosal barrier and confer gut health benefits. This study aims to investigate the implications of continuous LGH intervening in intestinal epithelium integrity and protective immunity against chemical dextran sodium sulfate (DSS)-induced colitis. Twelve male BALB/c mice were randomly arranged into four groups. The LGH/DSS group had results in bodyweight variance, epithelial cell density, and aberrancy score as good as the LGH group, and both were equivalent to the control group. LGH consumption effectively protects the distal intestinal epithelium by activating innate T lymphocytes. Meanwhile, T-cell subsets in subepithelial interspersion take a bystander role in these microenvironmental alterations. Under this stress, the cluster of differentiation 3 (CD3)^+^ T cells infiltrate the epithelium, while CD4^+^ T cells inversely appear in submucosal large lymphoid aggregates/isolated lymphoid follicles (ILFs) in which significant CD3^+^, CD4^+^, and CD8^+^ T-cell populations agglomerate. Moreover, forkhead box P3 (Foxp3) and interleukin 17 (IL-17) are observed in these ILFs. Agglomerated CD4^+^ T-cell lineages may have roles with proinflammatory T helper 17 cells and anti-inflammatory regulatory T cells in balancing responses to intraluminal antigens. Collectively, LGH administration may function in immune modulation to protect against DSS-induced inflammation.

## 1. Introduction

Diet directly impacts gut microbiota composition and diversity [1]. Gut microbial community changes may occur through species selection, via host communication; this can occur when specific digestive metabolites that modulate immune homeostasis [2] are produced (e.g., indoles, aryl hydrocarbon receptor ligands, short chain fatty acids, and polyamines) [3].

Different strategies, including prebiotics, probiotics, postbiotics, antibiotics, and fecal microbiota transplantation, have been discussed with the goal of gut microbiota regulation [4,5]. Among these strategies, prebiotic supplementation is usually considered a highly safe approach [5]. Prebiotics have been defined as substrates that are selectively utilized by those host microorganisms which confer health benefits [6]. Compared with probiotics, which are composed of live microorganisms, prebiotics are nonviable ingredients; they broadly include dietary fibers, polyphenols, and polyunsaturated fatty acids [7]. Among these prebiotics, dietary fibers are hydrophilic polysaccharides with an array of compounds. Their physicochemical properties vary by size, chemical structure, solubility, viscosity, and fermentability, each of which influence microbial competence [8].

The intestinal mucosal barrier is a complex structure, which functions interdependently on gut microbial symbiosis [3], epithelial homeostasis [9], and activation of gut-associated lymphoid tissues (GALT) [10]. GALT are composed of Peyer’s patches of the ileum and lymphoid aggregates/isolated lymphoid follicles (ILFs) of the colon [10]. ILFs are considered a primary core for regional intestinal immunity [11]. Immune tolerance of the mucosal barrier depends largely on the balance between T helper 17 (Th17) and regulatory T (Treg) cells [12,13], which differentiate from cluster of differentiation 4 (CD4)^+^ T lymphocytes at mature ILFs [12]. The barrier function is improved by the downregulation of low-grade mucosal immune activation, increasing the mucus layer and production of tight junction proteins [14]. Patients suffering from metabolic and gastrointestinal disorders are at a moderate-to-high risk of infection with novel severe acute respiratory syndrome coronavirus 2, indicating the direct implication of gut dysbiosis in the severity of the coronavirus disease 2019 [15].

Hemicelluloses such as popular glucomannan (also called ‘konjac’) are usually considered prebiotics [16] as they both undergo hydrophilic decomposition and are easily transported to the distal gut [17] for microbial fermentation [18]. Native konjac glucomannan has limited uses and has been chemically/physically/enzymatically modified to expand the range of functional properties, including biodegradable film, medical and pharmaceutical material, and encapsulation [17,19]. Herein, we used longish glucomannan hydrolysates (LGH) as an intervention in a murine model of dextran sodium sulfate (DSS)-induced colitis [20] to test a prebiotic strategy to managing gut bacteria. The specific distribution of innate T lymphocytes and the immune effects of LGH on gut health need to be better understood. Therefore, the aim of this study was to investigate the implications of continuous LGH intervening in intestinal epithelium integrity and protective immunity against induced inflammation.

## 2. Materials and Methods

### 2.1. Mouse Model of DSS-Induced Colitis and LGH Supplementation

Male BALB/c mice (6–8 weeks old) were purchased from the National Laboratory Animal Center (Taipei, Taiwan) and maintained in the Animal Research Center at Cathay General Hospital (Taipei, Taiwan), according to the regulations of the Institutional Animal Care and Use Committee of Cathay General Hospital (approval No. IACUC108-008). All animals were housed in plastic cages (3 or 4 mice/cage) under the following conditions: humidity (50 ± 10%), light (12/12 h light/dark cycle), and temperature (23 ± 2 °C). Mice were quarantined for seven days before being randomly assigned by body weight into one of four groups (see flowchart in Figure 1): (1) Control group (*n* = 3), normal chow diet and drinking water; (2) LGH group (*n* = 3), delivering LGH (1.5 mg/day; 10–40 kDa; PA1080503, Bioceutical Attainments Co., Ltd., Kaohsiung, Taiwan) by gavage feeding; (3) DSS group (*n* = 3), normal feeding plus 4% DSS (approximately 40 kDa; D5144, Tokyo Chemical Industry, Tokyo, Japan) in drinking water for colitis induction; (4) LGH/DSS group (*n* = 3), LGH with gavage feeding plus 4% DSS in drinking water.

### 2.2. Inflammation Characteristics of Colon Tissue in Mice with Induced Colitis

Following treatments, mice with induced inflammation were euthanized, and the abdominal cavity was opened. The colon was isolated and opened longitudinally, and the inflammation status was characterized through macroscopic, histological, and immunohistochemical (IHC) analyses [21]. Isolated colons were fixed, dehydrated, and embedded in paraffin according to standard histopathological techniques, and 5 μm sections were cut and transferred to slides. Histological examination of the sections involved hematoxylin and eosin (H&E) staining [22] and IHC analysis to detect inflammatory markers (for tumor necrosis factor-α (TNF-α), GTX15821, 1:100; GeneTex, Irvine, CA, USA; for interleukin-6 (IL-6), ab208113, 1:100; Abcam, Cambridge, UK), cytokines (forkhead box P3 (Foxp3), ab215206, 1:100 and IL-17, ab79056, 1:200; both from Abcam), and T-cell receptors (CD3, ab16669, 1:500; CD4, ab183685, 1:600; CD8, ab22378, 1:800; all from Abcam). Briefly, tissue section slides were immersed in Tris-EDTA buffer (10 mM Tris base, 1 mM EDTA solution, and 0.05% Tween 20; pH 9.0) and incubated for 20 min on a hot plate (95–99 °C) or boiled and then cooled to room temperature for 20 min. The tissue was then blocked using a blocking solution (VECTASTAIN Elite ABC kit; Vector Laboratories, Burlingame, CA, USA) for 2 h. After three washes in PBS buffer, sections were incubated for another 16 h at 4 °C with different proteins. This was followed by a 15 min incubation to block endogenous peroxidase by 0.3% H_2_O_2_, a 60 min incubation to retrieve target proteins using a biotin-labeled secondary antibody with appropriate development using a peroxidase substrate solution, and counterstaining each slide with hematoxylin. After dehydration and mounting, slides were examined by pathologists.

### 2.3. Immunohistochemical Evaluations of Cytokine Expressions

Proinflammatory cytokines IL-6 and TNF-α were evaluated in blinded fashion by clinical pathology consultants. Expression levels were estimated using a semiquantitative scale of grade I (signal intensity) and grade II (cell expansion based on the proportion of positive stain cells) [23] (see scoring scheme in Appendix A). Two grades were added, for a level range of 0–4 used to assess the expression of IL-6 or TNF-α.

### 2.4. Calculation of Epithelial Cell Density

Non-discriminated enterocyte nuclei were counted, and the lengths of corresponding basement membranes were measured in Figure 2 using ImageJ software (Bethesda, MD, USA), allowing the calculation of cell density (cell/μm).

### 2.5. Assessment of Inflammatory Scores and Analysis of Aberrancy Score

Inflammatory score (IS) was used to determine the degrees of damage and inflammation in histological (H&E-prepared) samples using the established grading system (see Appendix A) [24,25]. Inflammation was graded from score of 0–3 for inflammatory severity, inflammatory extent, epithelium regeneration, and crypt damage and then multiplied by the involvement of 1–4 (each IS range: 0–12). Four inflammatory scores were then summed as an aberrancy score (AS) ranging from 0 to 48 (listed in AS equation). Thirty sections were scored and analyzed. Blinded scoring was based on: (i) markers of severe inflammation including crypt abscesses; submucosal inflammation and ulceration; (ii) lamina propria leukocyte infiltration; (iii) epithelial hyperplasia and goblet cell depletion; and (iv) area of tissue affected [26]:AS = IS _inflammatory severity_ + IS _inflammatory extent_ + IS _epithelium regeneration_ + IS _crypt damage_
where IS = inflammation × involvement.

### 2.6. Statistical Analysis

Results are expressed as mean ± standard error when normally distributed, or as median (25–75%) otherwise. One-way ANOVA followed by Bonferroni post hoc test was performed to compare groups. Differences were considered statistically significant when *p* < 0.05. As adequate homogeneity was demonstrated, changes in the outcome measures from pretest to post-test and group differences in these changes were tested using the generalized estimating equation (GEE).

## 3. Results

### 3.1. Bodyweight Variance

Variance in bodyweight was used to evaluate the animals’ health status. We averaged weights within each group during the experimental period (28 days) (Figure 3A). To estimate the effects of LGH and DSS on bodyweight between days 21 (pretest) and 28 (posttest), GEE was used to compare between-groups differences, controlling for the pretest–post-test difference (Table 1). There was not a significant difference in pretest–post-test bodyweight variances between the LGH and LGH/DSS groups, but there was a significant difference in the DSS group. This may indicate that LGH protected the animals from DSS-induced damage.

### 3.2. Integrity of Colonic Epithelium

We then calculated the cell density of colonic epithelium from the relative cell numbers along the edge of the basement membrane. Briefly, cell density of the DSS group was the significantly lowest among the four groups (*p* < 0.001) (Figure 3B). However, the LGH/DSS group remained equivalent to the LGH group despite receiving DSS treatment. There was not a significant difference between the control and LGH administration (LGH and LGH/DSS) groups.

### 3.3. LGH Manipulates DSS-Induced Inflammatory Responses

Histological assessment was used to evaluate the inflammatory impacts (Table 2). Compared with the DSS group, LGH administration with/without DSS treatment led to low-level inflammation, reflected by all analyzed features (*p* < 0.001). As shown in Figure 3C, the AS for the DSS group differed markedly from the other groups (all *p* < 0.001). The groups receiving continuous LGH did not differ significantly from the control. The two critical proinflammatory cytokines (IL-6, TNF-α) of the colonic mucosa are described in Table 3 and Table 4. Their expressions were significantly reduced in the LGH/DSS group compared with the DSS group. Furthermore, IL-6 expression in the superficial mucosa of the LGH/DSS group was significantly lower compared with the control (*p* < 0.05) and higher to the LGH group (*p* < 0.001) in the submucosa, while TNF-α expression was not. IL-6 expression cells were spread sporadically across the submucosa, and certain cells were accumulated at lymphoid aggregates in the LGH/DSS group. We also observed many necrotic intestinal epithelial cells (IECs) and crypt abscesses in the DSS group (Figure 4). Referring to the LGH consumption groups, enterocytes had an intact shape. Nevertheless, these cell impairments with architectural erosion in the LGH/DSS group, distinct from necrosis with folliculate enlargement in the DSS group, were obviously shrunken. As shown in Table 2, the inflammatory score of epithelial regeneration or crypt damage in the LGH/DSS group was markedly better (i.e., at a lower level) compared with the DSS group. In fact, the regeneration of IECs and turnover of crypts were sufficient to maintain intestinal barrier homeostasis.

### 3.4. Lamina Propria T Lymphocyte Activation

T lymphocytes were detected with IHC CD3 and CD4 stained. T cells with CD3^+^ and/or CD4^+^ stains appeared irregularly in the lamina propria of the control group (Figure 5). When ongoing LGH was administered, activated T cells were interspersed among the subepithelium. However, CD3^+^ but not CD4^+^ T cells were confined to the lateral intercellular space of the epithelium and infiltrated the crypts in the LGH/DSS group (lower right panel in Figure 5A for CD3 and Figure 5B for CD4). In contrast, there were fewer in the DSS group (lower left panel in Figure 5A for CD3 and Figure 5B for CD4). Intriguingly, CD4^+^ T cells had extensive vanishing from the stroma of the lamina propria in the LGH/DSS group and were agglomerated in submucosal large lymphoid aggregates.

### 3.5. Lymphoid Aggregate Characteristics

The lymphoid aggregates/ILFs are normal GALT components. The presence and volume of mature lymphoid aggregates are related to the response to intraluminal antigens. There are B-cell germinal centers and T-cell outer follicles agglomerated in mature ILFs [10], where memory B cells present antigens to T cells [27]. Intramucosal small/immature lymphoid aggregates of the DSS group may represent a localized inflammatory response to mucosal damage (Figure 6A). In contrast, submucosal large lymphoid aggregates in the LGH/DSS group represent the activation of an adaptive immune response to intraluminal antigen. A relatively enhanced immune response was observed in the LGH/DSS group, predominantly submucosal large lymphoid aggregates, in which T-cell distributions had higher CD3^+^, CD4^+^, and CD8^+^ T-cell populations (Figure 6B). In addition, the Foxp3 and IL-17 expressions were observed to be spread in large quantities over the submucosa of the DSS group instead of agglomerated in the small lymphoid aggregates (Figure 6A). Conversely, Foxp3 and IL-17 were also highly expressed in submucosal large lymphoid aggregates of the LGH/DSS group (Figure 6B), showing an enhanced immune defense against inflammation and pathogens.

## 4. Discussion

LGH showed a protective effect on the colonic epithelium in DSS-induced colitis, helping to explain underlying prebiotic mechanism. LGH has a short, linear backbone and hydrophilic β-mannan-derived substrate intended to active the β-mannan-polysaccharide utilization loci (PUL) of gut commensal microflora for dominant competence [28]. β-mannan was selected for the reasons. Firstly, two common phyla of the gut microbiota, Firmicutes and Bacteroidetes, allow mannan-PUL activation, and the Actinobacteria phylum can assimilate the cross-fed metabolites to energy [29]. Secondly, the bacterial metabolites are butyrate-prominent, which is regarded as a beneficial nutrient for colonocytes [30]. Although the ‘longish’ saccharides were used with the expectation that they would be fermented by the distal gut microflora, whether LGH directly binds to the unanticipated enterocyte Toll-like receptors [31] to induce host defense activity remains unanswered.

The metabolic simulation of human gut microflora synchronizing the decomposition of β-mannan has been conducted in murine experiments [32,33]. β-mannan selectively promotes beneficial bacteria and a reduction in mucus degraders, exemplified by increased *Roseburia intestinalis* [32]. In mixed culture, *R. intestinalis* dominates the primary degradation in dietary polymeric β-mannan and shares the available β-mannan with *Bacteroides ovatus* [32]. Then, *Faecalibactzerium prausnitzii* coexisting with the primary degraders to ferment the primary metabolites results in syntrophic growth [33]. Certain species in the genera of *Bifidobacterium*, *Roseburia*, *Faecalibacterium*, and *Clostridium* are also efficient at assimilating communal primary metabolites to expand their own territories in a cascade effect [34].

Recently, dietary fibers were found to influence the human gut microbiota, inducing CD4^+^ T cell activation and the accumulation of co-stimulator-expressing CD8^+^ and CD4^+^ T cells in the tumor of late-stage melanoma patients responsive to anti-PD-1 immunotherapy [35]. While the relation between dietary fibers and immune responses remains blurry, the status of adaptive peripheral T-cell activation in the case is unavoidable correspondence with innate immune modulation of GALT. Another reported [36] indicated that the co-occurrence of tumor-associated CD8^+^ T cells and CD20^+^ B cells is associated with the formation of tertiary lymphoid structures in melanoma and improved immunotherapy and survival. Consistent with this, B cells have been shown to sustain inflammation and CD4^+^ and CD8^+^ T-cell numbers in the tumor microenvironment.

Continuous LGH administration in mice model leads to immune responses by innate T-cell activation, promoting epithelium integrity and protective immunity. ILFs in the LGH group contained few T lymphocytes, which were scattered, with CD3^+^ cell numbers like those of CD4^+^ cells, and far more than CD8^+^ cells. Murine ILF maturation depends on the formation of cryptopatches, whose development requires retinoic-acid-related orphan receptor gamma t (RORγt)^+^ c-kit^+^ innate lymphoid cells type 3 (ILC3s) and sensing intestinal aryl hydrocarbon receptor and microbiota-derived ligands [10]. Mature ILF contains a central B-cell cluster of up to 50–70% of total cells, which is surrounded by a ring of ILC3s (approximately 25% of total cells) [10], suggesting that ILC3s may be involved in the development, maintenance, and function of ILFs in mice. Generally, the lineage of ILC3s is capable of secreting IL-17A and IL-22 to regulate interaction with gut microbiota at the mucosal barrier surfaces and reacts with RORγt transcription factor to promote Th17 CD4^+^ T cell maturation [27]. Dramatically, ILC3s expresses the major histocompatibility complex class II (MHC II) to drive type-1 immunity that inhibits microbiota-specific effector CD4^+^ T cell responses, which is sensitive to anti-PD-1 immunotherapy in colon cancer [37]. In that study, an abundance of four microflora were successfully transferred from human fecal microbiota transplantation belonging to the *Bacteroides* genus and significantly correlated with anti-PD-1 responses [37]. Moreover, in murine colitis and colorectal cancer models, ILC3s exhibit increased phenotypic plasticity and transited to ex-ILC3s (RORγt^−^ ILC3s) or ILC1s driven by IL-23 and transforming growth factor β (TGF-β) in the induced lymphoid follicles, which are ILFs-like tissues termed ‘tertiary lymphoid structure’ [27]. The structure is induced by chronic inflammation in most organs independently of lymphoid tissue inducer cells, through the expression of LTα1β2 by B-cell, T cell, and NK-cell subsets [38]. Thus, the dysregulated ILCs do not protect from inflammation or tumorigenesis.

Our data suggest that both CD4^+^ T cell and CD8^+^ T cell populations were poorly agglomerated in the small lymphoid aggregates of the DSS group, indicating less ILC3-specific activity. In contrast, ILFs in the LGH/DSS group were larger and more multiple compared with the LGH group, indicating that protective immunity driven by LGH may be associated with the maturation of lymphoid aggregates, and the presence of large numbers of CD8^+^ T cell and CD4^+^ T cell populations with highly expressing Foxp3 and IL-17. Collectively, CD4^+^ T cells play roles by maintaining Th17 and Treg balance, which may enhance immune tolerance [13] of the mucosal barrier. Therefore, the enlarged ILFs may function in immunomodulation to protect against DSS-induced inflammation.

Two lamina propria T lymphocyte lineages are observed to be activated in the maintenance of intestinal homeostasis. Subepithelial CD3^+^ T cell and CD4^+^ T cell subsets had bystander roles in observing enterocytes in the LGH group and CD3^+^ T cells infiltrate the epithelium and crypts of the LGH/DSS group. The infiltrated cells were also stained by IHC IL-6 (Appendix A), whose IL-6 expression may help epithelium cell repairs. IL-6 is a pleiotropic cytokine able to prevent epithelial apoptosis during prolonged inflammation and produced by intraepithelial lymphocytes at the onset of an inflammatory injury for intestinal epithelial proliferation and wound repairs [39]. Additionally, IL-6 were moderate expression in submucosal ILFs, where IL-6 was an essential cytokine for early RORγt expressed by microbe-dependent Th17 cells and led to their differentiation [40]. CD4^+^ T cells in the LGH group were more widely distributed than in the control group. The distribution was similar to the IHC Foxp3 stained group (Appendix A), indicating LGH-inducing CD4^+^ Treg activities in the lamina propria. Moreover, large numbers of CD4^+^ T cells appeared in large lymphoid aggregates, which nearly disappeared in the lamina propria of the LGH/DSS group. This status suggests that a portion of CD4^+^ T cells in the ILFs had originated from lamina propria instead of peripheral naïve T lymphocytes. In contrast, IL-17 and Foxp3 were highly expressed in the submucosa of the DSS group, indicated by the dominance of peripheral T lymphocytes. Unless innate immunity engages in homeostatic maintenance, an overactive ‘cytokine storm’ response may develop [41]. Together, innate CD3^+^ and CD4^+^ lamina propria T lymphocytes display diverse routes to improve intestinal mucosal barrier.

Though our study can provide the immune responses in mice following LGH administration, there are still some issues that need to be noted. Limitations include the limited number of mice in each group and the exploratory nature of the immunity analyses. For example, the role of Toll-like receptors involved in the LGH-induced innate immunity was not assessed. Furthermore, we did not include the changes in fecal microbiota caused by LGH, which limits our findings to only prebiotic supplementation.

## 5. Conclusions

The LGH-based induction of innate T lymphocyte activities to improve intestinal mucosal barrier was observed. In this observational study, continuous LGH administration may have activated CD3^+^ and CD4^+^ lamina propria T lymphocyte lineages, allowing environmental alterations as a distinctive response. In conjunction, CD3^+^ T cells infiltrated the epithelium for cell repairs and CD4^+^ T cells agglomerated in the ILFs for immune modulation. These responses may have coordinated with ILC3 participation and deployment [27]. Future directions should target the differentiation of these relations to clarify the role of ILC3s between regional mucosal immunity and adaptive immunity driven by LGH. The prebiotic LGH has been shown to improve the integrity of intestinal epithelium and protective immunity. It may indicate the potential effect of LGH on the prophylactic or complementary treatment of colitis or colorectal cancer.

## Figures and Tables

**Figure 1 nutrients-14-02220-f001:**
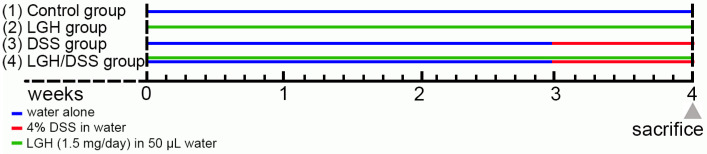
Timing of DSS induction of colitis and LGH administration. Mice were randomly divided into the groups as follows: (1) Control group (*n* = 3; blue line) with normal chow diet and drinking water; (2) LGH group (*n* = 3; green line) with LGH gavage feeding; (3) DSS group (*n* = 3) with 4% DSS (indicated by red line) in drinking water; (4) LGH/DSS group (*n* = 3) with LGH gavage feeding plus 4% DSS in drinking water. Gray triangle, end of day for sacrifice. DSS, dextran sodium sulfate; LGH, longish glucomannan hydrolysates.

**Figure 2 nutrients-14-02220-f002:**
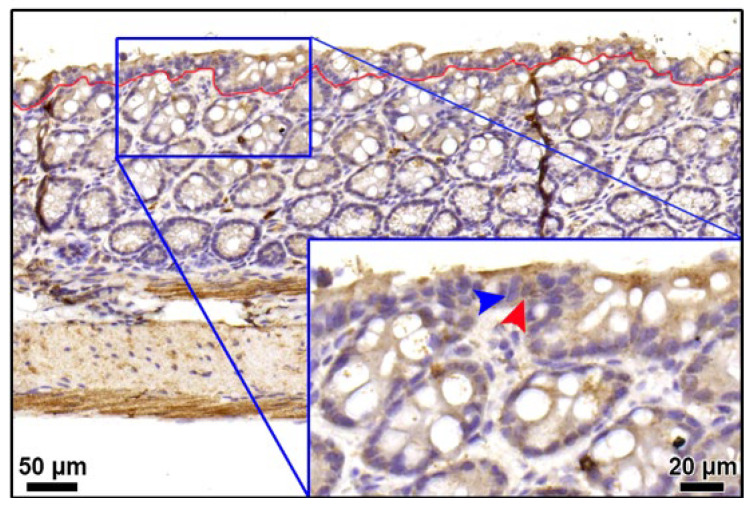
Representative image of TNF-α immunoreactivity for enterocyte nuclei. Immunohistochemical analysis stated the presence and location of TNF-α protein in colon specimen (scale bar, 50 µm). The superficial epithelium was the layer above the red curve. Inset showed the corresponding area at higher magnification (scale bar, 20 µm). Blue arrowhead, the TNF-α-negative enterocyte nucleus; red arrowhead, the TNF-α-positive immunocyte nucleus.

**Figure 3 nutrients-14-02220-f003:**
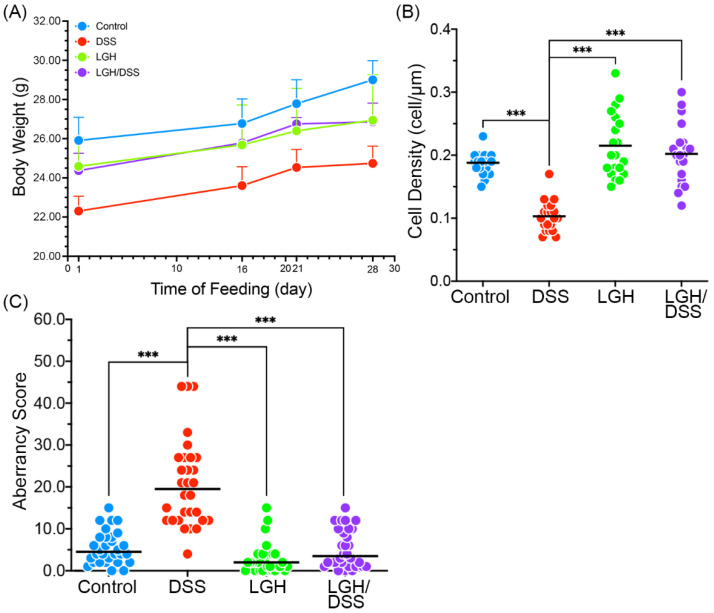
The analysis of body weight and colonic epithelium of mice following LGH administration. (**A**) Body weights of different groups. Error bars represented the positive SD. The averages of the cell density (**B**) and the aberrancy score (**C**) were calculated from 30 sessions for each group. Blue circles, data from the Control group; red circles, data from the DSS group; green circles, data from the LGH group; purple circles, data from the LGH/DSS group. DSS, dextran sodium sulfate; LGH, longish glucomannan hydrolysates. ***, *p* < 0.001.

**Figure 4 nutrients-14-02220-f004:**
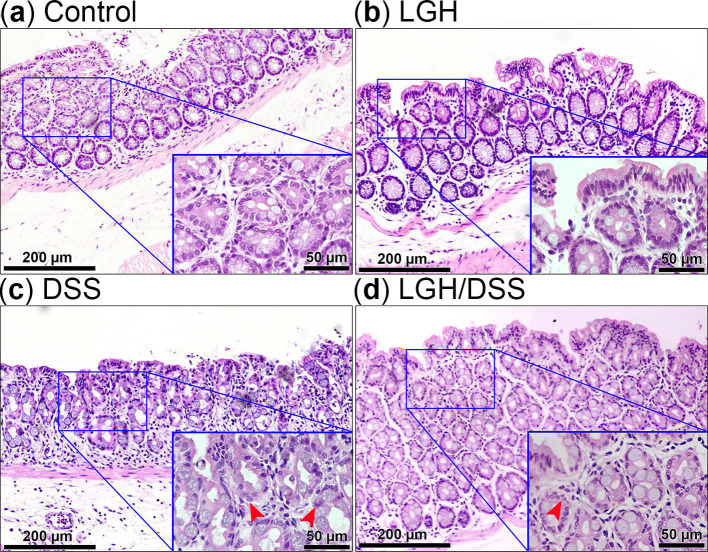
Representative histological staining from mouse colons of different groups. Colon tissue sections from control group (**a**), LGH group (**b**), DSS group (**c**), and LGH/DSS group (**d**) were stained with hematoxylin and eosin (H&E; scale bar, 200 µm). Each inset represented the corresponding area of H&E staining at higher magnification (scale bar, 50 µm). Red arrowheads representatively indicated the cells with folliculate enlargement in the DSS group and with architectural shrunken in the LGH/DSS group. DSS, dextran sodium sulfate; LGH, longish glucomannan hydrolysates.

**Figure 5 nutrients-14-02220-f005:**
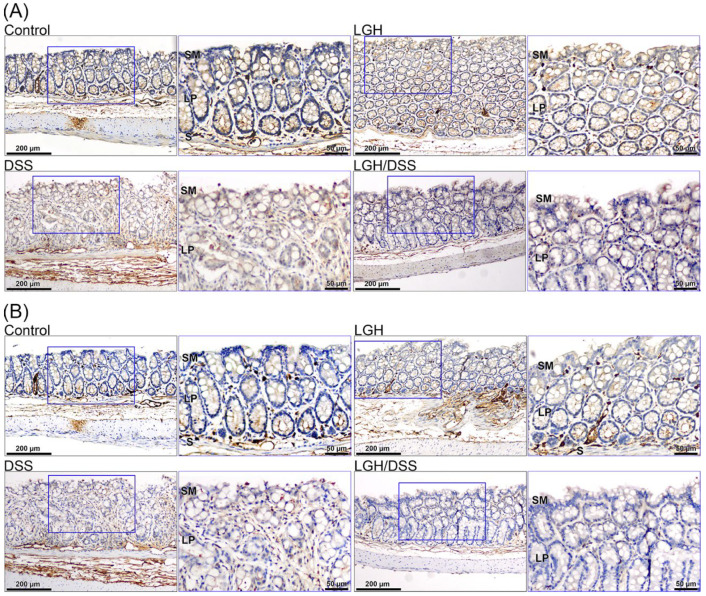
Representative immunohistochemical staining for the expression of CD3 and CD4 in the colons of different groups. Colon tissue sections from control group, LGH group, DSS group, and LGH/DSS group (scale bar, 200 µm) were stained with CD3 (**A**) and CD4 (**B**). Each inset represented the corresponding area (blue square) at higher magnification (scale bar, 50 µm). DSS, dextran sodium sulfate; LGH, longish glucomannan hydrolysates; SM, superficial mucosa; LP, lamina propria; S, submucosa.

**Figure 6 nutrients-14-02220-f006:**
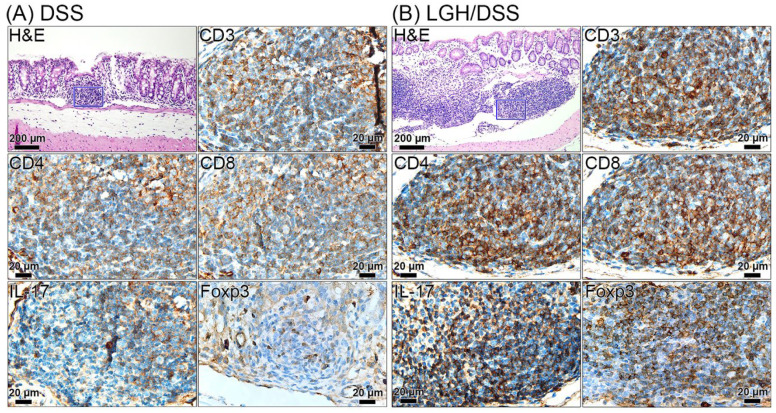
Representative immunohistochemical staining for the expression of CD3, CD4, CD8, IL-17, and Foxp3 in the lymphoid aggregates in colons of different groups. (**A**) DSS and (**B**) LGH/DSS. Representative images from serial tissue sections showed the target protein in the same region of lymphoid aggregates (inset, indicated by blue square in H&E staining sections of (**A**) and (**B**); scale bar, 200 µm). Each inset represented the target protein (CD3, CD4, CD8, IL-17, and Foxp3) at higher magnification (scale bar, 20 µm). DSS, dextran sodium sulfate; LGH, longish glucomannan hydrolysates.

**Table 1 nutrients-14-02220-t001:** GEE analysis of bodyweight between-groups differences.

Variables	β	SE	χ^2^	*p*
Time				
Pre-test	Ref			
Post-test	0.66	0.07	90.3	***
Group				
Control	Ref			
DSS	−3.26	0.63	26.7	***
LGH	−1.38	1.12	1.53	0.22
LGH/DSS	−1.03	0.48	4.60	*
Group *×* Time				
Control *×* Pre-test	Ref			
Control *×* Post-test	Ref			
DSS *×* Pre-test	Ref			
DSS *×* Post-test	−0.45	0.08	33.7	***
LGH *×* Pre-test	Ref			
LGH *×* Post-test	−0.12	0.12	1.01	0.32
LGH/DSS *×* Pre-test	Ref			
LGH/DSS *×* Post-test	−0.55	0.33	2.72	0.10
Intercept	27.8	0.46	3724	***

Results were obtained from generalized estimating equations (GEE) method. Model-based estimator with structure of AR1 correlation matrix was used. β, coefficient; SE, standard error; χ^2^, chi-squared test; DSS, dextran sodium sulfate; LGH, longish glucomannan hydrolysates; Ref, reference group; * *p* < 0.05, *** *p* < 0.001.

**Table 2 nutrients-14-02220-t002:** Multiple comparisons of histological assessment of colitis.

Feature	Group (G)	Inflammatory Score		Multiple Comparisons		
		Mean	SE	(G)–Control	(G)–LGH	(G)–DSS
Inflammatory	Control	0.6	0.2	0.0	0.2	−3.5 ***
severity	DSS	4.1	0.4	3.5 ***	3.7 ***	0.0
	LGH	0.4	0.2	−0.2	0.0	−3.7 ***
	LGH/DSS	0.8	0.2	0.2	0.4	−3.3 ***
Inflammatory	Control	0.9	0.3	0.0	0.5	−2.4 ***
extent	DSS	3.3	0.3	2.4 ***	2.9 ***	0.0
	LGH	0.4	0.2	−0.5	0.0	−2.9 ***
	LGH/DSS	1.2	0.3	0.3	0.8	−2.1 ***
Epithelium	Control	0.3	0.1	0.0	0.0	−2.9 ***
regeneration	DSS	3.2	0.5	2.9 ***	2.9 ***	0.0
	LGH	0.3	0.1	0.0	0.0	−2.9 ***
	LGH/DSS	0.2	0.1	−0.1	−0.1	−3.0 ***
Crypt	Control	0.4	0.1	0.0	0.2	−2.6 ***
damage	DSS	3.0	0.5	2.6 ***	2.8 ***	0.0
	LGH	0.2	0.1	−0.2	0.0	−2.8 ***
	LGH/DSS	0.2	0.1	−0.2	0.0	−2.8 ***

Multiple comparisons were analyzed by Dunnett T3 method if non-homogeneity of variances was present (all of features in this case). DSS, dextran sodium sulfate; LGH, longish glucomannan hydrolysates; SE, standard error; *** *p* < 0.001.

**Table 3 nutrients-14-02220-t003:** Multiple comparisons of IL-6 expression in colonic mucosa and submucosa.

Location	Group (G)	Grade		Multiple Comparisons		
		Mean	SE	G–Control	G–LGH	G–DSS
Superficial	Control	0.8	0.1	0.0	0.2	−1.7 ***
mucosa (SM)	DSS	2.5	0.2	1.7 ***	1.9 ***	0.0
	LGH	0.6	0.1	−0.2	0.0	−1.9 ***
	LGH/DSS	0.3	0.1	−0.5 *	−0.3	−2.2 ***
Lamina	Control	1.7	0.1	0.0	0.3	−0.5
propria (LP)	DSS	2.2	0.2	0.5	0.8 ***	0.0
	LGH	1.4	0.1	−0.3	0.0	−0.8 ***
	LGH/DSS	1.4	0.1	−0.3	0.0	−0.8 ***
Submucosa (S)	Control	2.0	0.2	0.0	0.5	−1.3 ***
	DSS	3.3	0.1	1.3 ***	0.8 ***	0.0
	LGH	1.5	0.2	−0.5	0.0	−0.8 ***
	LGH/DSS	2.3	0.1	0.3	0.8 ***	−1.0 ***

Multiple comparisons were analyzed by Scheffe’s method if homogeneity of variances was present (submucosa in this case), and the others were analyzed by Dunnett T3 method instead. Scoring results were stated on Appendix A. DSS, dextran sodium sulfate; LGH, longish glucomannan hydrolysates; SE, standard error; * *p* < 0.05, and *** *p* < 0.001.

**Table 4 nutrients-14-02220-t004:** Multiple comparisons for TNF-α expression in colonic mucosa and submucosa.

Location	Group (G)	Grade		Multiple Comparisons		
		Mean	SE	G–Control	G–LGH	G–DSS
Superficial	Control	1.2	0.1	0.0	−0.4	−2.1 ***
mucosa (SM)	DSS	3.3	0.2	2.1 ***	1.7 ***	0.0
	LGH	1.6	0.1	0.4	0.0	−1.7 ***
	LGH/DSS	1.4	0.2	0.2	−0.2	−1.9 ***
Lamina	Control	1.1	0.1	0.0	−0.2	−2.0 ***
propria (LP)	DSS	3.1	0.1	2.0 ***	1.7***	0.0
	LGH	1.3	0.1	0.2	0.0	−1.7 ***
	LGH/DSS	1.3	0.1	0.2	0.0	−1.7 ***
Submucosa (S)	Control	0.2	0.1	0.0	0.1	−2.4 ***
	DSS	2.6	0.2	2.4 ***	2.5 ***	0.0
	LGH	0.1	0.1	−0.1	0.0	−2.5 ***
	LGH/DSS	0.7	0.1	0.5	0.6	−1.9 ***

Multiple comparisons were analyzed by Scheffe’s method if homogeneity of variances was present (mucosal surface; basal mucosa in this case), and the others were analyzed by Dunnett T3 method instead. Scoring results were stated on Appendix A. DSS, dextran sodium sulfate; LGH, longish glucomannan hydrolysates; SE, standard error; *** *p* < 0.001.

## Data Availability

The data and material that support the findings of this study are available from the corresponding author upon reasonable request. Anonymized data will be shared by request from any qualified investigator.

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
