# Peer review of "Intestinal Mucosal Barrier Improvement with Prebiotics: Histological Evaluation of Longish Glucomannan Hydrolysates-Induced Innate T Lymphocyte Activities in Mice"

_nutrients, 2022, doi:10.3390/nu14112220_

Round 1
Reviewer 1 Report
Dear Authors,
I revised your manuscript entitled ''Intestinal Mucosal Barrier Improvement with Prebiotics: Longish Glucomannan Hydrolysates-Induced Innate T Lymphocyte Activities'' which presents a relevant topic for today's research interests. Still, some changes are required to improve the quality of your work. In the attached document you may find specific comments that could amend the soundness of your work.
Bests regards,
Reviewer

Author Response
- Please reconstruct the Abstract section by highlighting the main results of your work.
Response: We appreciate the reviewer’s comment. We have revised the Abstract by highlighting our main results. We hope that this revised abstract can meet the requirements of “Nutrients”.
- Please give more details on gut mucosal barrier integrity. Please use the following references:
-Simon, Elemer, et al. "Probiotics, prebiotics, and synbiotics: Implications and beneficial effects against irritable bowel syndrome." Nutrients 13.6 (2021): 2112.
-Vodnar, Dan-Cristian, et al. "Coronavirus disease (COVID-19) caused by (sars-cov-2) infections: A real challenge for human gut microbiota." Frontiers in cellular and infection microbiology (2020): 786.
Response: We appreciate the reviewer’s comment. These two references (#14 and #15, respectively) are added and stated in the section of Introduction (lines 114-119, on Page 2).
- More information describing glucomannan is needed. Please check the following references:
-Zhang, Cui, and Feng-qing Yang. "Konjac glucomannan, a promising polysaccharide for OCDDS." Carbohydrate polymers 104 (2014): 175-181.
-Zhu, Fan. "Modifications of konjac glucomannan for diverse applications." Food chemistry 256 (2018): 419-426.
-Devaraj, Ramya Devi, Chagam Koteswara Reddy, and Baojun Xu. "Health-promoting effects of konjac glucomannan and its practical applications: A critical review." International journal of biological macromolecules 126 (2019): 273-281.
Response: We appreciate the reviewer’s comment. We introduce more properties of glucomannan in the 4th paragraph of the section of Introduction and add the “Carbohydrate polymers 104 (2014): 175-181” as reference #17 and “Food chemistry 256 (2018): 419-426” as reference #19 in lines 122-125, on Page 2.
- Please indicate the standard reference for histological examination.
Response: Thanks for reviewer’s reminding. We look for two references (#21 and #22, in the revised version) to indicate the standard techniques for histological examinations in the section of Materials and Methods (lines 165 and 168, on Page 3).
- Please define the abbreviations at their first use. Please do the same for all situations.
Response: We appreciate the reviewer’s carefulness, and the abbreviations are defined when they are first used.
- Images need a higher resolution.
Response: We appreciate the reviewer’s comment. All the images in this article are 300 dpi. However, we insert the insets in Figure 5 with higher magnification to demonstrate where the positive signals of CD3 or CD4 exist.
- Please use ''significant'' instead of ''significance''.
Response: Thanks for reviewer’s carefulness. The inappropriate ''significance'' in line 230 on page 4 is replaced with ''significant''.
- ''Lamina'' instead of ''Laminar''
Response: Thanks for reviewer’s carefulness. All the incorrect ''laminar'' are replaced with the correct ''lamina'' in the revised version.
- Please increase the picture resolution, if possible.
Response: We appreciate the reviewer’s comment. As replying to the comment #6, all the images in this article are 300 dpi.
Reviewer 2 Report
This is a well writiten manuscript on a well conducted study of prebiotic effects. I only have few comments to the authors:
1. add study design to the title
2. abstract must be improved. What was the aim? Setting? Study period?
3. line 70 ...therefore we aimed...
4. start discussion with your main findings
5. add limitation section
6. add to conclusion what is the impact of your results on possible clinical use and possible future studies
Author Response
- add study design to the title
Response: We appreciate the reviewer’s comment. We have changed the title to “Intestinal Mucosal Barrier Improvement with Prebiotics: Histological Evaluation of Longish Glucomannan Hydrolysates-Induced Innate T Lymphocyte Activities in Mice”. We hope this revised title can convey our study design.
- abstract must be improved. What was the aim? Setting? Study period?
Response: As we replying to the comment #1 of reviewer 1, we have revised the Abstract to meet the requirements of “Nutrients”.
- line 70 ...therefore we aimed...
Response: Thanks for reviewer’s comment. This statement is inappropriate. In combination with other comments of reviewers, we have changed these sentences to “The specific distribution of innate T lymphocytes and the immune effects of LGH on gut health need to be better understood. Therefore, the aim of this study was to investigate the implications of continuous LGH intervening in intestinal epithelium integrity and protective immunity against induced inflammation” (lines 127-131, on Page 2).
- start discussion with your main findings
Response: Thanks for reviewer’s comment. We add one sentence “LGH showed a protective effect on the colonic epithelium in DSS-induced colitis, helping to explain underlying prebiotic mechanism” to start the section of Discussion.
- add limitation section
Response: Thanks for reviewer’s comment. We add the limitation at the end of the section of Discussion. Briefly, we summary the shortcomings and limitations of this study.
- add to conclusion what is the impact of your results on possible clinical use and possible future studies
Response: Thanks for reviewer’s comment. We have revised the last sentences in the section of Conclusion to “Future directions should target differentiation of these relations to clarify the role of ILC3s between regional mucosal immunity and adaptive immunity driven by LGH. The prebiotic LGH has been shown to improve the integrity of intestinal epithelium and protective immunity. It may indicate the potential effect of LGH on the prophylactic or complementary treatment of colitis or colorectal cancer” on Page 12.